# Use of Ultrasound and Ki–67 Proliferation Index to Predict Breast Cancer Tumor Response to Neoadjuvant Endocrine Therapy

**DOI:** 10.3390/healthcare11030417

**Published:** 2023-02-01

**Authors:** Sean C. Liebscher, Lyndsey J. Kilgore, Onalisa Winblad, Nika Gloyeske, Kelsey Larson, Christa Balanoff, Lauren Nye, Anne O’Dea, Priyanka Sharma, Bruce Kimler, Qamar Khan, Jamie Wagner

**Affiliations:** 1Department of Surgery, University of Kansas Medical Center, Kansas City, KS 66160, USA; 2Department of Radiology, University of Kansas Medical Center, Kansas City, KS 66160, USA; 3Department of Pathology and Laboratory Medicine, University of Kansas Medical Center, Kansas City, KS 66160, USA; 4Department of Internal Medicine, University of Kansas Medical Center, Kansas City, KS 66160, USA; 5Department of Radiation Oncology, University of Kansas Medical Center, Kansas City, KS 66160, USA

**Keywords:** breast cancer, neoadjuvant endocrine therapy, response to treatment

## Abstract

Background: Prediction of tumor shrinkage and pattern of treatment response following neoadjuvant endocrine therapy (NET) for estrogen receptor positive (ER+), Her2 negative (Her2–) breast cancers have had limited assessment. We examined if ultrasound (US) and Ki–67 could predict the pathologic response to treatment with NET and how the pattern of response may impact surgical planning. Methods: A total of 103 postmenopausal women with ER+, HER2– breast cancer enrolled on the FELINE trial had Ki–67 obtained at baseline, day 14, and surgical pathology. A total of 70 patients had an US at baseline and at the end of treatment (EOT). A total of 48 patients had residual tumor bed cellularity (RTBC) assessed. The US response was defined as complete response (CR), partial response (PR), stable disease (SD), and progressive disease (PD). CR or PR on imaging and ≤70% residual tumor bed cellularity (RTBC) defined a contracted response pattern. Results: A decrease in Ki–67 at day 14 was not predictive of EOT US response or RTBC. A contracted response pattern was identified in one patient with CR and in sixteen patients (33%) with PR on US. Although 26 patients (54%) had SD on imaging, 22 (85%) had RTBC ≤70%, suggesting a non-contracted response pattern of the tumor bed. The remaining four (15%) with SD and five with PD had no response. Conclusion: Ki–67 does not predict a change in tumor size or RTBC. NET does not uniformly result in a contracted response pattern of the tumor bed. Caution should be taken when using NET for the purpose of downstaging tumor size or converting borderline mastectomy/lumpectomy patients.

## 1. Introduction

Breast cancer is one of the most commonly diagnosed cancer types, with an estimated 257,850 cases diagnosed in the US in 2022 [1]. Thirteen percent of all women in the US will be diagnosed with breast cancer in their lifetime, and three percent will die from the disease [1]. Surgery and chemotherapy have traditionally been the mainstays of treatment; however, neoadjuvant endocrine therapy (NET) has shown promise in recent years as a treatment option for patients with estrogen receptor positive (ER+) and human epidermal growth factor 2-negative (Her2–) breast cancers. NET is an efficacious treatment method that is well tolerated by patients [2,3,4,5,6,7,8,9,10,11,12]. When compared with neoadjuvant chemotherapy, NET has improved patient tolerance and compliance, as well as cost-effectiveness, both in terms of treatment cost and in the utilization of the healthcare system [13,14].

Most studies evaluating NET, however, have focused on oncologic outcomes, drug selection/duration, and identifying prognostic markers. There is limited data describing how NET affects surgical planning. Initial studies of NET demonstrated that some patients who were initially mastectomy candidates were able to undergo breast conservation therapy (BCT) following treatment [15,16]. Since that time, however, assessment of the prediction of tumor size reduction and pattern of treatment response following NET has been limited. There is limited data on predictive markers that could suggest whether a patient’s response to NET therapy will be robust enough to alter surgical planning.

Several studies have evaluated the use of the Ki–67 proliferation index of a tumor during treatment as an independent prognostic tool for oncologic outcomes such as recurrence, death following recurrence, and relapse-free survival [4,17,18,19]. Whether the Ki–67 proliferation index can offer insight into the pathologic response pattern to treatment with NET, however, has had limited evaluation. There is a possibility that, while patients may have a robust response to NET in terms of the Ki–67 proliferation index, the tumor may not uniformly contract in size. For example, patients could have a decrease in residual tumor bed cellularity yet not have a change in maximum tumor bed dimensions. This would suggest a response to NET that correlates with the previously demonstrated oncologic outcomes, as the number of overall tumor cells present in the tumor bed has decreased but has not affected the overall size of the tumor. This phenomenon, where the number of viable tumor cells in a tumor bed is reduced by treatment without affecting the overall tumor dimensions, will be referred to as a non-contracted response pattern. This would contrast with a contracted response pattern, a treatment response where the overall number of tumor cells decreased simultaneously with a decrease in the size of the tumor bed. Non-contracted response patterns are important to study as we assess NET’s utility in converting patients from mastectomy to BCT [20,21]. It is conceivable that patients could have favorable oncologic outcomes following treatment with NET that result in no change in overall tumor size.

The incidence of contracted versus non-contracted response patterns after NET is unknown and has the potential to affect surgical planning. Considering this, we sought to determine two things. First, if Ki–67 expression in combination with ultrasound (US) has a predictive value in terms of the pathologic response to treatment, and second, to evaluate the frequency of contracted versus non-contracted tumor response patterns following NET to allow for better informed surgical planning for patients undergoing NET.

## 2. Methods 

We evaluated postmenopausal women with ER+, HER2– breast cancer who were enrolled on the FELINE trial. The FELINE trial is a prospective, randomized, multi-center, and placebo-controlled trial comparing NET using letrozole with and without a CDK4/6 inhibitor [22]. We conducted a retrospective cohort study based on the participants using data collected from the FELINE trial. As part of the FELINE trial, patients had the Ki–67 proliferation index of their tumor obtained at baseline, day 14 of treatment, and on final surgical pathology. Patients who withdrew consent prior to completing the trial and patients who did not have complete Ki–67 proliferation index data were excluded. Patients were also excluded if their US at both baseline and end of treatment (EOT) were not available for assessment (Figure 1).

Baseline US, EOT US, pathologic tumor size, and residual tumor bed cellularity (RTBC) were compared to determine the US correlation with the final pathologic tumor size and pattern of treatment response. US response to treatment was defined as complete response (CR) with resolution of the target lesion on EOT US, partial response (PR) with ≥30% reduction in the largest dimension, stable disease (SD) <30% reduction or <20% increase in the largest dimension, or progressive disease (PD) with ≥20% increase in the largest dimension.

The Ki–67 proliferation index was calculated by pathologists specializing in breast cancer. It was determined by comparing the number of Ki–67-positive tumor cells with the total number of tumor cells present.

The pattern of treatment response was determined using a combination of US imaging response size as well as RTBC. Previous studies have suggested 5 year survival rates >74% for tumors with ≤70% RTBC while tumors with >70% RTBC had 5 year survival rates <60% [10]. Therefore, we selected 70% RTBC as a cutoff and defined a contracted response pattern as patients with CR or PR on imaging and ≤70% RTBC as their tumors decreased in both dimension and cellularity. A non-contracted response pattern was defined as SD on imaging with ≤70% RTBC on final surgical pathology, as these tumors decreased in cellularity with only minimal change in the imaging size dimensions. Patients with PD or SD and >70% RTBC were defined as not responding to treatment. A subset analysis was performed on the patients that had RTBC assessed on their final surgical pathology.

Statistical analysis was performed with SPSS for Windows version 27 (IBM Corp.). Due to the number of participants and non-normal distributions for several variables, non-parametric approaches were used. Chi-squared or Fisher’s exact tests were used for categorical variables, and Mann–Whitney or Kruskal–Wallis tests were used for continuous variables. For correlation between continuous variables, the non-parametric Spearman’s rho test and linear regression analysis were used. All test results were reported as two-sided, uncorrected for multiple comparisons, and with *p* < 0.05 being considered statistically significant.

## 3. Results

One hundred and twenty postmenopausal women with ER+, HER2–breast cancer enrolled on the FELINE trial were evaluated [22,23]. We excluded 14 of the initial 120 patients who withdrew consent prior to completing the trial and three patients who did not have complete Ki–67 proliferation index data obtained. Of the 103 women with complete Ki–67 data available, 70 had an US at baseline and end of treatment (EOT) available for assessment. Forty-eight patients had residual tumor bed cellularity (RTBC) assessed on the final surgical pathology.

A decrease in Ki–67 from baseline to day 14 strongly correlated with a sustained decrease in Ki–67 at the time of surgery (*p* < 0.001, Figure 2). Despite this robust association, changes in Ki–67 were not predictive of the tumor response seen on US (Figure 2) or of RTBC. The tumor size on the final surgical pathology did correlate with the size on the EOT US (*p* = 0.024). Of the 70 patients with pre-treatment and EOT US available, two (3%) had a CR, twenty-eight (40%) had a PR, thirty-three (47%) had SD, and seven (10%) had PD.

Analysis of the 48 patients with RTBC assessed demonstrated a single patient with CR on US and 10% RTBC on surgical pathology. All 16 patients (33%) that had a PR on US also had a RTBC ≤ 70%. Twenty-six patients (54%) had SD on imaging, and twenty-two (85%) of these patients had a RTBC ≤ 70%. The remaining four patients (15%) with SD had >70% RTBC. Five patients with PD had no response to treatment. This data was then used to determine the response pattern. Of the 48 patients who had RTBC assessed, a contracted response pattern was seen in 17 patients (35.4%): one patient with CR and sixteen patients with PR, all of whom had RTBC ≤ 70%. Although 26 patients had SD on imaging, 22 (84.6%) had a RTBC ≤ 70%, consistent with a non-contracted response pattern. The remaining four patients with SD and five patients with PD had no pathologic response to treatment.

## 4. Discussion

As the Ki–67 proliferation index has demonstrated efficacy in predicting oncologic outcomes following chemotherapy [4,17,18,19], we hypothesized that it may have some utility in predicting changes in tumor size following treatment with NET. Interestingly, there was a robust correlation in the 103 patients whose Ki–67 was assessed between the change in Ki–67 from baseline to day 14 of treatment and the change in Ki–67 from baseline to final surgical pathology. Furthermore, a decrease in Ki–67 observed on day 14 strongly correlated with a sustained decrease in Ki–67 seen at the time of surgery (Figure 2).

The change in Ki–67 proliferation index from baseline to day 14 was compared to the change in tumor size from the pre-treatment US to the EOT US to determine if the change in Ki–67 at day 14 could be used to predict a change in tumor size over the course of treatment. Unfortunately, despite the observed correlation in decreased Ki–67 levels, there was no correlation between change in tumor size over the course of treatment and Ki–67 levels (Figure 3). These findings suggest that a change in Ki–67 does not predict a decrease in pathologic tumor size. This is similar to the findings of Liu et al., who found that, for patients treated with NAC, Ki–67 was only predictive of a decrease in tumor size if pre-treatment Ki–67 levels were ≥14% [24].

The size of the tumor on EOT US was then compared to the size of the tumor on surgical pathology to confirm the use of US measurements as a reasonable estimation for pathologic tumor size. The linear correlation we found between tumor size on surgical pathology and tumor size on EOT US demonstrates that EOT US is a reasonable tool to estimate pathologic tumor size (Figure 4). This reaffirms that US is an appropriate choice for EOT imaging following NET to assess treatment response and aid in surgical planning [25]. While this conclusion makes sense, it also serves as an internal control that our various methods of measuring tumor size, namely US and final pathological size, are in agreement. It also suggests that the use of tumor size on US, in combination with RTBC, to determine the difference in tumor response patterns is a reasonable proxy for the actual pathologic tumor size.

Response to treatment was determined by changes in tumor size on US as well as the RTBC on surgical pathology. Based on US imaging, approximately 43% (30/70 patients) of patients showed a response to treatment; 2 had CR and 28 had PR. The remaining 57% (40/70 patients) of patients had either SD or PD. This suggests that the use of NET solely to decrease tumor size is a technique that should be employed with caution, as less than half of patients will have a treatment response pattern that will lead to a substantial decrease in the greatest dimension of the tumor.

Only a minority of patients (43%) on NET demonstrated response to therapy based on size, which contrasts with the favorable oncologic outcomes associated with NET. This discrepancy may be explained by the pathologic response pattern observed following NET, which is presumed to be different than the response pattern observed in patients undergoing neoadjuvant chemotherapy (NAC). For patients receiving NAC, there is typically a decrease in the overall size of the tumor in response to therapy [26]. We defined this type of pathologic response pattern, where the greatest dimension of the tumor decreases over the course of therapy, as a contracted response. We hypothesized that, for patients undergoing NET, there would be a portion of patients who would respond in a contracted fashion. However, we also suspected there would be a portion of patients whose tumors did not significantly change in size despite a response to treatment. We called this type of response, where the response to treatment was without a change in tumor size, a non-contracted response.

For the purposes of our study, we defined a contracted response to treatment as a decrease in the greatest dimension of the tumor bed from pre-treatment to EOT US. A non-contracted response was defined as SD observed on US coupled with a RTBC ≤ 70%. SD with RTBC ≤ 70% would suggest there was a robust response to NET, as the overall cellularity of the tumor decreased, but this response did not result in a significant decrease of the greatest dimension of the tumor.

Of the 70 patients with US imaging available, 48 had the residual cellularity of their pathologic tumor bed evaluated and were available for analysis. For the 17 patients who had either a CR (n = 1) or PR (n = 16) on US, all of them had RTBC ≤ 70%. This implied a pattern of contracted response, as evidenced both by the change in overall tumor size, as well as the decrease in tumor bed cellularity. This response pattern is similar to the response one would traditionally expect to occur following NAC [26]. For the 26 patients with SD on US, 85% (n = 22) were found to have RTBC ≤ 70%, which indicated a non-contracted response to treatment. If a traditional, contracted response pattern alone is assumed to represent the response rate to NET, it would appear that less than half of the tumors (43%) responded to treatment. This would not match the favorable oncologic outcomes associated with NET. However, when patients with non-contracted response patterns (i.e., those with SD on US but ≤70% RTBC) are included, 89% of tumors demonstrate response to treatment. This level of treatment response is much more consistent with reported NET outcomes [2,3,4].

While this high probability of a positive response to NET is consistent with previously reported results, the possibility that this response could be either a contracted or non-contracted response highlights the fact that caution must be used if trying to employ NET to convert patients from mastectomy candidates to breast conserving therapy candidates [5]. While patients have a high likelihood of a successful response to NET in terms of overall response to treatment (89% of patients), this translates to a decrease in the amount of breast tissue that must be excised for an appropriate oncologic resection less than half of the time (43% of patients). This is an important consideration when discussing the risks and benefits of neoadjuvant treatment options with patients. While there are advantages to NET in terms of cost-effectiveness, patient tolerance, and compliance, patients will need to consider the possibility that their tumor size may not decrease with NET. For patients who prefer mastectomy regardless, or for those whose tumors are an appropriate size for BCT preoperatively, the possibility of a minimal change in overall tumor size may be trivial. For those patients who desire BCT but whose tumors are too large for BCT, the increased possibility of decreasing the size of the tumor bed and becoming a lumpectomy candidate with NAC could be a reason to select NAC over NET. This will be an important piece of information for surgeons, medical oncologists, and patients to discuss as they are selecting an appropriate, individualized treatment plan.

Next steps for further investigation should include evaluation of other imaging modalities in addition to US, such as MRI or mammography, to determine their prognostic value in NET. Additionally, it would be beneficial to study the relationship between NET and changes in surgical management to determine how frequently mastectomy candidates are converted to breast conservation therapy candidates with NET. While our data would imply that there would likely not be as many patients converted from mastectomy to lumpectomy as compared with NAC, it would be interesting to explore this further and determine what clinical impact NET has on surgical planning. Additionally, the phenomenon of a non-contracted response pattern may also explain why a decrease in Ki–67 does not correlate with a change in tumor size and could be an area of further exploration. It is possible that the response to Ki–67 does, in fact, predict response to treatment, but only if additional measurements other than the size of the tumor, such as RTBC, are considered.

There are some limitations to this study. There is a lack of demographic data in our dataset that prevents us from making any comments about the heterogeneity of our study population. We have no reason to suspect any particular bias in the demographics of the study participants, as the data used for this study was a retrospective analysis of the data from a prospective, randomized, multi-center, and placebo-controlled trial. However, we have no way to determine this conclusively as this data was not available for examination, and the results should be interpreted in light of this. Second, to define a contracted versus non-contracted response pattern, we needed to select a cutoff value for RTBC. While the value of 70% residual cellularity was selected prior to data evaluation based on our literature review, a higher or lower cutoff could impact the results [23].

## 5. Conclusions

During NET, the change in Ki–67 expression does not predict a change in tumor size on final surgical pathology and therefore is not a useful indicator when trying to predict response to treatment, at least when overall size is the marker of response to treatment. This could be because there is a significant portion of patients who have a robust response to NET but do not have a change in the size of the tumor bed. When evaluating response to NET going forward, it will be important to keep this population of patients in mind: those who exhibit a non-contracted response pattern. Patients with non-contracted response patterns could significantly change the outcomes of future studies if overall tumor bed size is utilized as the sole marker to determine response to treatment.

Additionally, more than half of the time, NET results in a non-contracted response pattern of the tumor bed. While this response pattern should not affect the overall oncologic outcomes, caution should be taken when using NET if one of the goals is decreasing tumor size for either improved cosmesis or converting mastectomy candidates to lumpectomy candidates.

## Figures and Tables

**Figure 1 healthcare-11-00417-f001:**
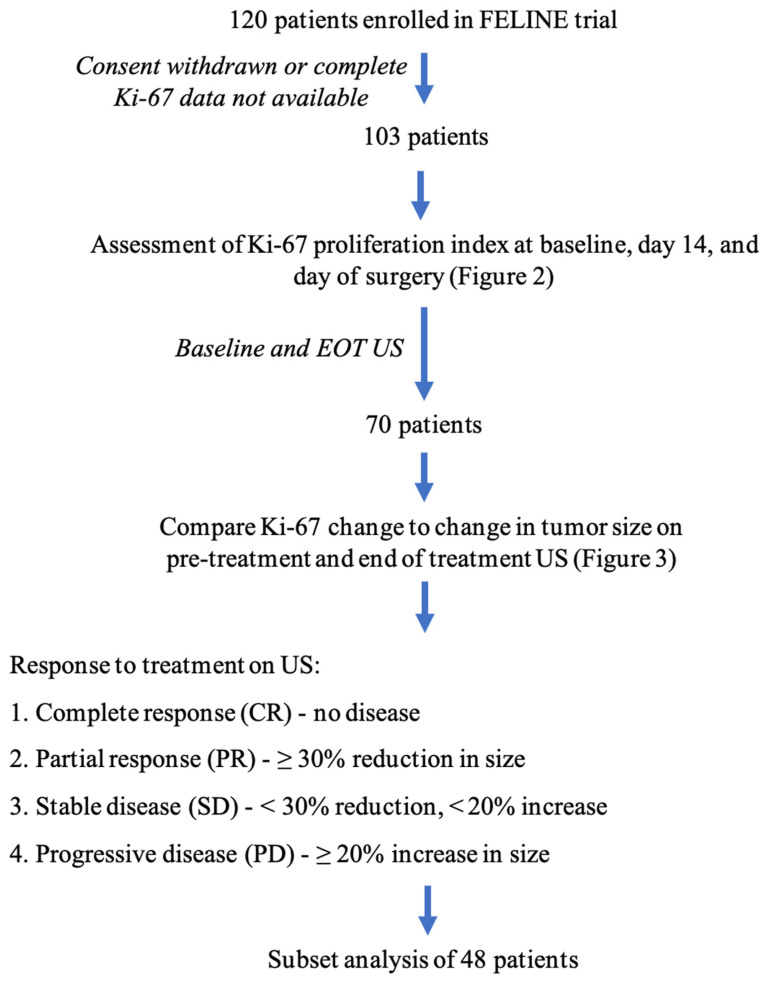
Graphical depiction of study methods. EOT = End of treatment; US = Ultrasound.

**Figure 2 healthcare-11-00417-f002:**
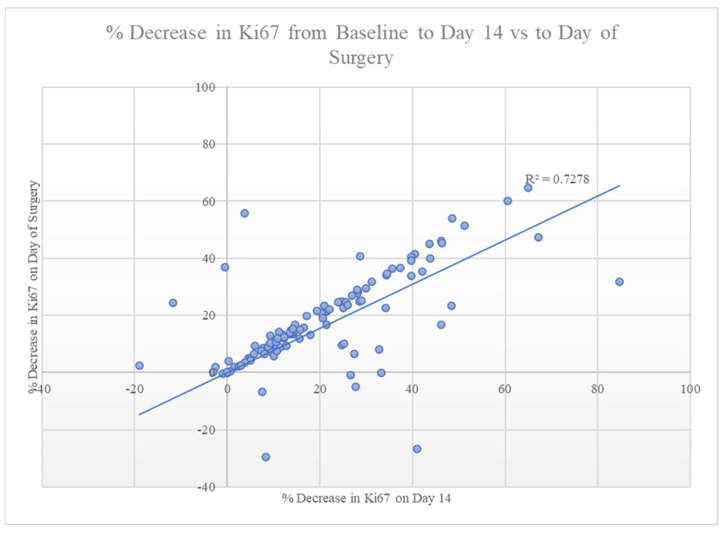
Correlation between the change in Ki–67 from baseline to day 14 versus baseline to the day of surgery, demonstrating a robust correlation between the two measurements (*p* < 0.001).

**Figure 3 healthcare-11-00417-f003:**
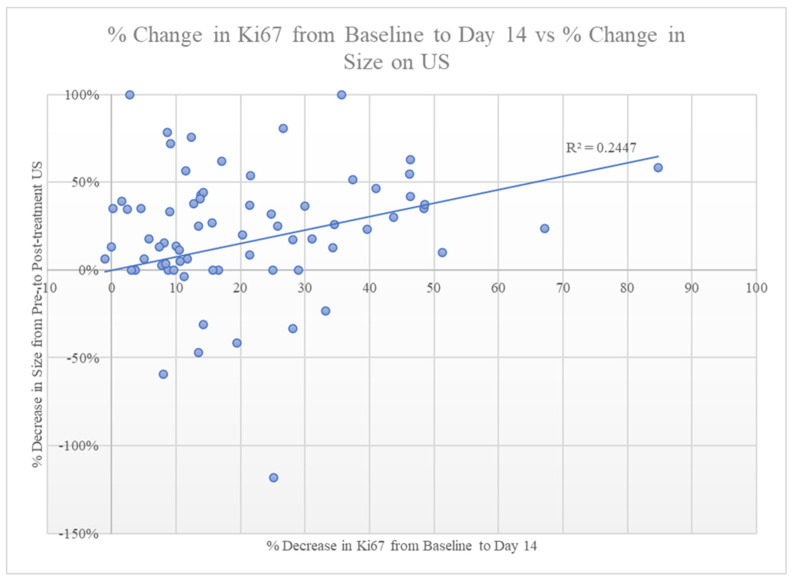
Comparison of the percentage decrease in the Ki–67 proliferation index from baseline to day 14 versus the percentage decrease in size seen on baseline US to EOT US, demonstrating no correlation between the two measurements.

**Figure 4 healthcare-11-00417-f004:**
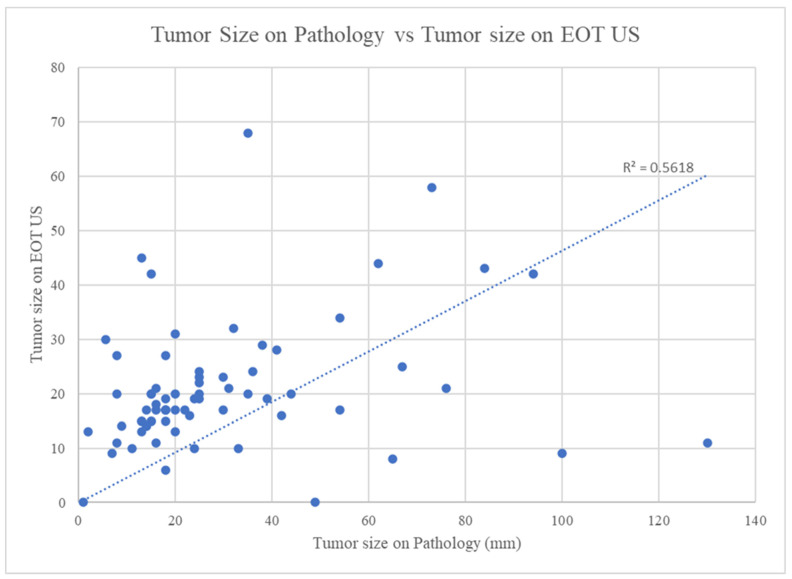
Comparison of the greatest dimension of tumor size on pathology (mm) to tumor size as determined by the greatest dimension on EOT US (mm). Tumor size on pathology correlates with tumor size on EOT US (*p* = 0.024). EOT = End of treatment; US = Ultrasound.

## Data Availability

The data presented in this study are available on request from the corresponding author. The data are not publicly available as the FELINE study data is not yet published.

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
