# Peer review of "Use of Ultrasound and Ki–67 Proliferation Index to Predict Breast Cancer Tumor Response to Neoadjuvant Endocrine Therapy"

_healthcare, 2023, doi:10.3390/healthcare11030417_

Round 1
Reviewer 1 Report
This work entitled “Use of Ultrasound and Ki-67 Proliferation Index to Predict 2 Breast Cancer Tumor Response to Neoadjuvant Endocrine Therapy” by Liebscher et al., They have reported ultrasound (US) and Ki-67 predict pathologic response to treatment with NET and how the pattern of response may impact surgical planning. In the present study, they concluded that Ki-67 does not predict a change in tumor size or RTBC and NET does not uniformly result in a contracted response pattern of the tumor bed. It is professionally written, well presented, and as such no major comments but there are some minor comments:-
(1). Introduction is kind of very brief. Authors should elaborate more to understand the topic.
(2). In the method section, the authors didn’t discuss how they assess the Ki-67 proliferative index.
(3). Please check the grammar of the research article.
Reviewer 2 Report
It's an exciting study contributing to an important topic. However, I have major comments before it gets published:
- The introduction needs to be extended, starting with the breast cancer definition and prevalences and screening, and here are related references
https://doi.org/10.3390/ijerph18010263
https://doi.org/10.1186/s12905-021-01543-7
and at the end of the introduction highlight the rationale of conducting this study and then link it with the your study objectives
- Under the methodology please include a subsection for study design and explain in detail the study design with the rationale for using this design.
- Under the methodology, please include another subsection for the sample size and sampling, explaining how you calculated the sample size and the method you have used for sampling to select your study sample.
- Under the methodology please include another subsection for statistical analysis and explain in detail the statistical tests you have used and why.
- In the results section, please include the descriptive analysis results of the socio-demographic characteristics of your sample.
- at the end of the discussion please discuss the theoretical and practical implications of your research.
- at the end of the discussion please discuss the strength and limitations of your research and your recommendations for future studies.
- In conclusion, it's important to reflect on your study results, theoretical and practical implications, and recommendations.
Round 2
Reviewer 2 Report
The authors didn't implement all comments rides during the first round to improve the manuscript and the manuscript is still very superficial and does not fit the rigor of this journal. Therefore, I am recommended the authors improve it again according to the comments and resubmit it again.
